# Examining patterns of multimorbidity, polypharmacy and risk of adverse drug reactions in chronic obstructive pulmonary disease: a cross-sectional UK Biobank study

Peter Hanlon,[1] Barbara I Nicholl,[1] Bhautesh Dinesh Jani,[1] Ross McQueenie,[1] Duncan Lee,[2] Katie I Gallacher,[1] Frances S Mair[1]

[1]General Practice and Primary Care, Institute of Health and Wellbeing, University of Glasgow, Glasgow, UK
[2]School of Mathematics and Statistics, University of Glasgow, Glasgow, UK

**Correspondence to**
Dr Frances S Mair;
frances.mair@glasgow.ac.uk

## ABSTRACT

**Objective** This study aims: (1) to describe the pattern and extent of multimorbidity and polypharmacy in UK Biobank participants with chronic obstructive pulmonary disease (COPD) and (2) to identify which comorbidities are associated with increased risk of adverse drug reactions (ADRs) resulting from polypharmacy.

**Design** Cross-sectional.

**Setting** Community cohort.

**Participants** UK Biobank participants comparing self-reported COPD (n=8317) with no COPD (n=494 323).

**Outcomes** Multimorbidity (≥4 conditions) and polypharmacy (≥5 medications) in participants with COPD versus those without. Risk of ADRs (taking ≥3 medications associated with falls, constipation, urinary retention, central nervous system (CNS) depression, bleeding or renal injury) in relation to the presence of COPD and individual comorbidities.

**Results** Multimorbidity was more common in participants with COPD than those without (17% vs 4%). Polypharmacy was highly prevalent (52% with COPD taking ≥5 medications vs 18% in those without COPD). Adjusting for age, sex and socioeconomic status, those with COPD were significantly more likely than those without to be prescribed ≥3 medications contributing to falls (OR 2.27, 95% CI 2.13 to 2.42), constipation (OR 3.42, 95% CI 3.10 to 3.77), urinary retention (OR 3.38, 95% CI 2.94 to 3.87), CNS depression (OR 3.75, 95% CI 3.31 to 4.25), bleeding (OR 4.61, 95% CI 3.35 to 6.19) and renal injury (OR 2.22, 95% CI 1.86 to 2.62). Concomitant cardiovascular disease was associated with the greatest risk of taking ≥3 medications associated with falls/renal injury. Concomitant mental health conditions were most strongly associated with medications linked with CNS depression/urinary retention/bleeding.

**Conclusions** Multimorbidity is common in COPD and associated with high levels of polypharmacy. Co-prescription of drugs with various ADRs is common. Future research should examine the effects on healthcare outcomes of co-prescribing multiple drugs with similar potential ADRs. Clinical guidelines should emphasise assessment of multimorbidity and ADR risk.

## Strengths and limitations of this study

► This paper assesses multimorbidity, polypharmacy and risk of adverse drug reactions (ADRs) in UK Biobank participants with self-reported chronic obstructive pulmonary disease (COPD) compared with those without COPD.

► Baseline variables from the UK Biobank assessment centre were used to adjust for potential confounders.

► Cumulative risk of common ADRs was quantified by identifying UK Biobank participants taking ≥3 medications associated with similar ADRs.

► Analyses were repeated using a subgroup of participants with spirometry data confirming airflow obstruction.

► Medication and comorbidity data rely on participant self-report and may thus be susceptible to bias or inaccuracy.

## BACKGROUND

In people with chronic obstructive pulmonary disease (COPD), multimorbidity (the presence of ≥2 long-term conditions (LTCs)) is highly prevalent.[1–4] A recent meta-analysis of 29 datasets demonstrated that those with COPD are significantly more likely to be diagnosed with a range of cardiovascular comorbidities than those without COPD (we will use the term comorbidity when referring to specific conditions in addition to COPD and multimorbidity to refer to the presence of ≥2 LTCs).[5] Other LTCs with known increased prevalence in COPD include obesity,[6] depression,[7–10] gastro-oesophageal reflux disease,[11–13] osteoporosis[14–16] and lung cancer.[17 18] Each of these conditions has been associated with poorer health-related outcomes in COPD when compared with those with no comorbidity.[19–30] The overall burden of multimorbidity also impacts

prognosis in COPD, for example higher number of comorbidities is associated with higher risk of mortality,[31] and higher burden of morbidity assessed using the Charlson index and the COPD-specific comorbidity test is associated with higher risk of all-cause and respiratory-specific mortality.[32 33] The importance of considering the impact of multimorbidity in the management of LTCs is increasingly recognised, particularly in the context of an ageing society in which the prevalence of multimorbidity is growing.[34] However, an immature evidence base means that disease-specific guidelines often lack specific recommendations with respect to multimorbidity.[35] The prevalence and prognostic significance of multimorbidity in COPD make it a potentially useful exemplar condition in which to consider the specific implications of different patterns of multimorbidity. Polypharmacy is one such implication.

Multimorbidity in the general population is associated with polypharmacy (often defined as concomitant use of ≥5 or ≥10 pharmacological agents).[36] Polypharmacy has been associated with increased risk of adverse drug reactions (ADRs)[37–39] and potentially preventable hospital admissions, particularly in the elderly.[40 41] This is particularly pertinent in an ageing society, in which a rising prevalence of polypharmacy has been observed.[34 37] It has been demonstrated that diagnosis of COPD is associated with increased risk of polypharmacy.[42 43] This is, in large measure, due to the high burden of extrapulmonary comorbidities.[44] However, little is known about the risk of ADRs in the context of multimorbidity in COPD.

Given the well-established burden of multimorbidity and polypharmacy in COPD, it is likely that those with COPD are at increased risk of ADRs resulting from polypharmacy. Previous analyses have not focused on the risk of specific ADRs, or assessed which LTCs increase this risk, instead reporting overall counts of prescribed medication. Data collected for the UK Biobank cohort offers an opportunity to assess how multimorbidity in COPD relates to polypharmacy and to assess the prevalence of co-prescription of medications with similar ADRs.

This paper aims:

► to describe the pattern and extent of multimorbidity and polypharmacy in UK Biobank participants with COPD;
► to identify which LTCs in people with COPD are associated with increased risk of ADRs resulting from polypharmacy.

## METHODS
### Data collection
The UK Biobank is a large, population cohort that recruited voluntary participants from throughout the UK. Between 2006 and 2010, UK Biobank recruited 502 640 participants aged 37–73. Participants underwent baseline assessments at one of 22 assessment centres throughout England, Scotland and Wales. Sociodemographic and lifestyle details were recorded using touchscreen questionnaires. Townsend scores were derived from participant postcodes to provide an area-based measure of socioeconomic deprivation. Self-reported LTCs, prescribed and over-the-counter medications, smoking status (current, previous or never) and frequency of alcohol intake (never/special occasions only, 1–3 times a month, at least once a week) were recorded from a touchscreen questionnaire and subsequent verbal interview with a study nurse. Physical activity was self-reported based on a questionnaire administered in the UK Biobank assessment centre.[45] We classified the responses into: none (no physical activity in the last 4 weeks), low (light ' do-it-yourself (DIY)' activity only in the last 4 weeks), medium (heavy DIY and/or walking for pleasure and/or other exercises in the last 4 weeks) and high (strenuous sports in the last 4 weeks).

Study centre staff also collected physical measures including height and weight (to calculate body mass index; BMI) and spirometry. Spirometry was performed using a Vitalograph Pneumotrac 6800. Individual reasons for contraindications to attempting spirometry were not recorded but, according to protocol, these included chest infection in the last month, history of collapsed lung and heart attack or surgery in the past 3 months. Full details of the Biobank spirometry protocol are available at their website (https://biobank.ctsu.ox.ac.uk/crystal/docs/Spirometry.pdf). In brief, participants were allowed up to three attempts to provide two reproducible spirometry measurements. Where the reproducibility of the first two was deemed acceptable (<5% variation in both forced expiratory volume in 1 s (FEV1) and forced vital capacity (FVC)), a third measurement was not performed. All values were recorded along with any error messages generated. As per the American Thoracic Society/European Respiratory Society (ATS/ERS) end-of-test criteria, we interpreted as valid any measurement with no error message or if 'user accepted' was specified.[46] No post-bronchodilator measurements were recorded, which deviates from the ATS/ERS guidelines and the Global Initiative for Chronic Obstructive Lung Disease (GOLD) guidelines for COPD.[47 48]

Participants provided full informed consent to participate in UK Biobank.

### Defining COPD
Participants reporting to have been diagnosed with COPD, chronic bronchitis or emphysema at the nurse-led interview were coded as having 'self-reported COPD'.

Due to the potential inaccuracies of using self-reported diagnoses, we identified a subset of those with self-reported COPD who met an adaptation of the GOLD spirometry criteria for COPD.[48] This subset, referred to as 'GOLD COPD', was used as a sensitivity analysis for self-reported COPD and to stratify findings by severity of airflow obstruction. For participants with self-report COPD and valid spirometry measurements, we calculated the ratio of FEV1 to FVC using the highest measurement for each participant meeting the ATS/ERS end-of-test criteria.[46] Those with a FEV1/FVC ratio <0.7 were classed as having

an obstructive deficit and thus meeting the GOLD diagnostic criteria for COPD. We used the Hankinson equation,[49] based on recorded age, sex and height, to calculate predicted FEV1 values for each participant. Those with GOLD COPD were classified on the basis of their best available FEV1 measurement as having mild (>80% predicted FEV1), moderate (50%–80% predicted FEV1), or severe (<50% predicted FEV1) airflow obstruction in line with the GOLD COPD guidelines.[48]

### Defining long-term conditions and medications

All LTCs were defined by self-report. The list of included LTCs was taken from a list of 42 morbidities originally established for a large multimorbidity epidemiological study in Scotland, through systematic review, the Quality and Outcomes Framework, NHS Scotland and an expert panel,[34] and subsequently amended for UK Biobank.[50] The inclusion of 'other painful conditions' comprised LTCs in which pain is a predominant feature (particularly as this is likely to influence medication use). It should be noted that such a list is not exhaustive, but intended to cover common conditions frequently requiring prescription of analgesics (eg, osteoarthritis, back pain, headaches, etc). Morbidities were categorised for the purposes of this analysis into cardiovascular disease, gastrointestinal disease, mental health conditions, cancer and painful conditions/inflammatory arthropathies (comprising the list of 'other painful conditions' mentioned above, plus connective tissue diseases). Full details of conditions comprising each category can be found in online supplementary appendix 1.

Medication data were collected by self-report. Medications were coded by mechanism of action according to the British National Formulary (BNF) (eg, ACE inhibitors, beta-blockers, calcium channel blockers, etc).[51] For some situations where more than one medication with a similar mechanism of action may be commonly co-prescribed (eg, aspirin and clopidogrel, both antiplatelets) these were coded separately. A complete list of the medications coded within each class can be found in online supplementary appendix 2.

We defined those at risk of specific ADRs as anyone on ≥3 medications with similar potential ADRs, based on information provided in the *Scottish Government Model of Care Polypharmacy Working Group: Polypharmacy Guidance.*[52] This guideline cross-tabulates commonly prescribed medications with common ADRs to help identify those at cumulative risk of ADRs. This document groups common medications by similar potential ADRs. While this list is not all-inclusive, and the cut-off value of ≥3 medications is arbitrary, this does allow an estimation of the cumulative risk of specific ADRs. We identified six potential ADRs (falls/fractures, constipation, urinary retention, central nervous system (CNS) depression, bleeding and renal injury) for which the proportion of participants taking ≥3 associated medications could be assessed. It should be noted that several of these events (eg, falls/fractures, CNS depression) are often multifactorial and medication may

be a contributing factor rather than a definitive cause. As the guideline acknowledges, however, these are clinical events of which the risk is increased by taking multiple associated medications.

### Statistical analysis

All analyses were planned prior to inspection of the data.

#### Baseline variables

Comparisons were made between participants with self-reported COPD and the rest of the cohort (who did not report COPD). Age, sex, smoking status, deprivation (Townsend score), BMI, physical activity and frequency of alcohol intake were compared using $\chi^2$ test for categorical variables, $\chi^2$ test for trend for ordinal variables and Mann-Whitney U test for continuous variables. Total number of morbidities, prevalence of specific morbidities, number of self-reported prescribed medications and proportion of participants taking each class of medication (online supplementary appendix 2) were also compared between those with self-reported COPD and the rest of the cohort. All comparisons were repeated comparing participants with GOLD COPD only with those without COPD, stratified by severity of airflow obstruction.

#### Multimorbidity and polypharmacy

Logistic regression analyses were used to compare participants with self-reported COPD and those without COPD. ORs and 95% CIs were calculated for:
► the presence of cardiovascular disease, cancer, gastrointestinal disease, mental health conditions and painful conditions/inflammatory arthropathies;
► the presence of ≥4 morbidities (excluding COPD);
► the use of ≥5 and ≥10, medications (two separate models).

Models were initially adjusted for age, sex and socioeconomic deprivation (model 1), then adjusted for the addition of smoking status, alcohol frequency, BMI and physical activity (model 2). These analyses were repeated comparing those with GOLD COPD only to those without COPD.

#### Risk of ADRs

For each potential ADR (falls/fractures, constipation, urinary retention, CNS depression, bleeding and renal injury) participants taking ≥3 medications associated with that ADR were identified. The following comparisons were then made:
► Unadjusted percentages at risk of each ADR were calculated for participants without COPD, with self-reported COPD and with self-reported COPD plus each category of LTC (cardiovascular disease, cancer, gastrointestinal disease, mental health conditions and painful conditions/inflammatory arthropathies) to give an impression of the ADR risk in COPD and identify LTCs in those with COPD that may increase this risk.
► ORs of being at risk of each ADR were calculated comparing those with self-reported COPD to those

without COPD adjusting for age, sex and socioeconomic deprivation (model 1) and for age, sex, socioeconomic deprivation, smoking status, alcohol frequency, BMI and physical activity (model 2).

► ORs of being at risk of each ADR were calculated comparing those with and without self-reported COPD in each LTC category to (ie, participants with cardiovascular disease alone compared with participants with cardiovascular disease plus COPD, etc). This was intended to identify whether specific patterns of multimorbidity in COPD are associated with increased ADR risk. Adjustment for a wide range of potential confounders was not appropriate in these models due to the smaller number of participants in each model.

Each analysis was repeated comparing GOLD COPD only to those without COPD. Less than 3% of participants (with or without COPD) had missing data for potential confounding variables (table 1). Those with missing data were excluded from adjusted analyses. Spirometry data were missing for 3591 participants with self-report COPD (43%), hence the use of the GOLD COPD subset as a sensitivity analysis.

All analyses were performed using R statistical software (V.3.3.1).

## RESULTS
At the time of recruitment, 8317 out of 502 619 participants reported having COPD (1.7%) and are referred to here as the self-report COPD group. Of those who self-reported COPD, 4726 (57%) had valid spirometry measurements. Spirometry was contraindicated or not available in 2507 of those with self-reported COPD. Spirometry measurements did not meet the ATS/ESR end-of-test criteria in 1084 participants.[46] Of those with valid spirometry, 2620 (55%) met the GOLD criteria for airflow obstruction (399 (15%) mild, 1409 (54%) moderate, 812 (31%) severe, see figure 1) and are referred to here as GOLD COPD.

### Baseline variables
Table 1 describes and compares the characteristics of those with and without COPD in UK Biobank. Participants with COPD (both self-report and GOLD) were significantly older, more socioeconomically deprived and less physically active. A higher proportion of those with COPD were male, obese and had a history of smoking.

### Multimorbidity and polypharmacy
Prevalence of each category of comorbidity was higher in those with COPD than without (table 2). After controlling for age, sex and socioeconomic status, those with self-reported COPD were significantly more likely than those without to have each category of LTC examined: cardiovascular disease (OR 1.45; 95% CI 1.39 to 1.52), cancer (OR 1.29, 95% CI 1.2 to

1.39), gastrointestinal disease (OR 1.76, 95% CI 1.67 to 1.86), mental health conditions (OR 2.02, 95% CI 1.89 to 2.15) and painful conditions (OR 1.54, 95% CI 1.46 to 1.62). Results for GOLD COPD also suggested higher likelihood of each LTC compared with those without COPD, although the ORs were lower and results for cancer not statistically significant (online supplementary appendix 3). Results were similar after adjusting for additional confounders (smoking status, alcohol frequency, BMI and physical activity) with the exception of cardiovascular disease in GOLD COPD, which was no longer significantly associated (OR 1.08, 95% CI 0.99 to 1.18) (online supplementary appendix 3).

Morbidity counts (excluding COPD) and counts of prescribed medication are shown in table 2 comparing those with COPD, stratified by severity of airflow obstruction, with those without. Those with COPD had higher numbers of LTCs and more prescribed medications than those without. There was a trend towards more prescribed medications in those with greater severity of airway obstruction. After controlling for age, sex and socioeconomic status, those with self-report COPD were more likely to report ≥4 LTCs (OR 3.49, 95% CI 3.28 to 3.71), ≥5 medications (OR 3.85, 95% CI 3.68 to 4.03) and ≥10 medications (OR 5.72, 95% CI 5.36 to 6.10) than those without COPD. Results were similar for GOLD COPD and remained statistically significant after adjusting for smoking status, alcohol frequency, BMI and physical activity (online supplementary appendix 3).

### ADR risk
Counts and percentages of participants taking specific medications are shown in online supplementary appendix 4. Participants with COPD (self-report and GOLD) were more likely that those without COPD to be prescribed drugs across a range of disease areas, reflecting the range of LTCs present among those with COPD. The percentages of participants within each category (no COPD, COPD and COPD with specific LTCs) taking ≥3 medications associated with a similar ADR is shown in figure 2. For each category of ADR a higher proportion of participants with COPD reported taking ≥3 associated medications than those without COPD. This increased further with multimorbidity. Participants with COPD plus cardiovascular disease had the highest percentage taking ≥3 medications with a risk of falls or renal injury. Participants with COPD plus mental health conditions had the highest percentages taking ≥3 medications with a risk of constipation, CNS depression or bleeding.

After adjusting for age, sex and socioeconomic deprivation, those with self-report COPD remained more likely to be taking ≥3 medications in each category than those without COPD. These findings remained statistically significant after adjusting for smoking status, alcohol frequency, BMI and physical activity (table 3). Findings were similar for GOLD COPD however, after adjusting for additional potentially confounding variables, results for bleeding risk

**Table 1** Baseline variables

| Characteristic | No COPD, n=494323 | | COPD (self-report), n=8317 | | | GOLD COPD, n=2620 | | |
|---|---|---|---|---|---|---|---|---|
| | Count | % | Count | % | P value | Count | % | P value |
| Sex | | | | | <0.001 | | | <0.001 |
| Male | 224906 | 45.5 | 4268 | 51.3 | | 1426 | 54.4 | |
| Female | 269417 | 54.5 | 4049 | 48.7 | | 1194 | 45.6 | |
| Age | | | | | <0.001 | | | <0.001 |
| | Median: 58 | | Median: 62 | | | Median: 63 | | |
| | IQR: 50–63 | | IQR: 57–66 | | | IQR: 59–66 | | |
| Ethnicity | | | | | <0.001 | | | <0.001 |
| White | 464770 | 94.5 | 8052 | 97.3 | | 2620 | 100 | |
| Other | 26821 | 5.4 | 219 | 2.6 | | 0 | 0 | |
| Missing | 2732 | | 46 | | | 0 | 0 | |
| Socioeconomic deprivation quintile | | | | | <0.001 | | | <0.001 |
| 1 (least deprived) | 99672 | 20.2 | 1015 | 12.2 | | 309 | 11.8 | |
| 2 | 98977 | 20.0 | 1142 | 13.7 | | 362 | 13.8 | |
| 3 | 99013 | 20.1 | 1399 | 16.8 | | 440 | 16.8 | |
| 4 | 98660 | 20.0 | 1735 | 20.9 | | 580 | 22.2 | |
| 5 (most deprived) | 98385 | 19.7 | 3015 | 36.3 | | 926 | 35.4 | |
| Missing | 616 | | 11 | | | 3 | | |
| Smoking status | | | | | <0.001 | | | <0.001 |
| Current | 50817 | 10.3 | 2172 | 26.3 | | 833 | 32.2 | |
| Previous | 169015 | 34.4 | 4083 | 49.5 | | 1398 | 54.0 | |
| Never | 271602 | 55.3 | 1999 | 24.2 | | 360 | 13.9 | |
| Missing | 2889 | | 63 | | | 29 | | |
| Alcohol frequency | | | | | <0.001 | | | <0.001 |
| Daily | 100070 | 20.3 | 1720 | 20.7 | | 618 | 23.6 | |
| 3–4 times/week | 114058 | 23.1 | 1404 | 16.9 | | 475 | 18.2 | |
| 1–2 times/week | 127459 | 25.9 | 1863 | 22.5 | | 561 | 21.5 | |
| 1–3 times/month | 54979 | 11.2 | 894 | 10.8 | | 289 | 11.1 | |
| Occasional | 56707 | 11.5 | 1322 | 15.9 | | 387 | 14.8 | |
| Never | 39569 | 8.0 | 1092 | 13.2 | | 284 | | |
| Missing | 1481 | | 22 | | | 6 | 10.9 | |
| BMI | | | | | <0.001 | | | <0.001 |
| <18.5 | 2478 | 0.5 | 148 | 1.8 | | 56 | 2.2 | |
| 18.5–24.9 | 155282 | 31.8 | 2185 | 26.8 | | 829 | 31.9 | |
| 25.0–29.9 | 211102 | 43.2 | 3165 | 38.9 | | 1049 | 40.4 | |
| >30 | 119813 | 24.5 | 2647 | 32.5 | | 665 | 25.6 | |
| Missing | 5648 | | 172 | | | 21 | | |
| Physical activity | | | | | <0.001 | | | <0.001 |
| High | 49827 | 10.6 | 250 | 3.1 | | 70 | 2.7 | |
| Medium | 387766 | 79.6 | 5838 | 72.0 | | 1902 | 73.3 | |
| Low | 18354 | 3.8 | 589 | 7.3 | | 203 | 7.8 | |
| None | 31425 | 6.4 | 1433 | 17.7 | | 421 | 16.2 | |
| Missing | 6951 | | 207 | | | 24 | | |
| FEV1 (% predicted) | | | | | <0.001 | | | <0.001 |
| >80 | 272109 | 78.0 | 1853 | 39.2 | | 399 | 15.2 | |
| 50–79 | 71727 | 20.6 | 2022 | 42.8 | | 1409 | 53.8 | |

| | No COPD, n=494 323 | | COPD (self-report), n=8317 | | | GOLD COPD, n=2620 | | |
|---|---|---|---|---|---|---|---|---|
| **Characteristic** | **Count** | **%** | **Count** | **%** | **P value** | **Count** | **%** | **P value** |
| <50 | 4841 | 1.4 | 851 | 18.0 | | 812 | 31.0 | |
| Missing | 145 646 | | 3591 | | | 1061 | | |

BMI, body mass index; COPD, chronic obstructive pulmonary disease; FEV1, forced expiratory volume in 1 s; GOLD, Global Initiative for Chronic Obstructive Lung Disease.

were not statistically significant in this sensitivity analysis (table 3).

Finally, each category of ADR risk was assessed in a subgroup analysis for each category of LTC (cardiovascular, GI, cancer, mental health and painful conditions/inflammatory arthropathies) comparing those with and without COPD (eg, participants with cardiovascular disease plus COPD compared with participants with cardiovascular disease alone, etc). These models were adjusted for age, sex and socioeconomic status only. Within each category of LTC, those with self-reported COPD were more likely to be at risk of each ADR than those without COPD (online supplementary appendix 3). Not all results were statistically significant when using GOLD COPD (online supplementary appendix 3).

## DISCUSSION
### Summary of main findings
Multimorbidity and polypharmacy in COPD were common among UK Biobank participants. The presence of multimorbidity was highly prevalent in those with COPD (85%). More than half (52%) reported polypharmacy (≥5 medications) and 15% reported ≥10 medications. The prevalence of cardiovascular disease, as well as the degree of polypharmacy, was higher among those with more severe airflow obstruction.

For the first time, our data demonstrate that those with COPD were more likely than those without to be prescribed multiple medications (≥3) with similar ADRs. Those with COPD plus cardiovascular disease were most likely to be taking multiple medications associated with increased risk of falls or renal injury, while

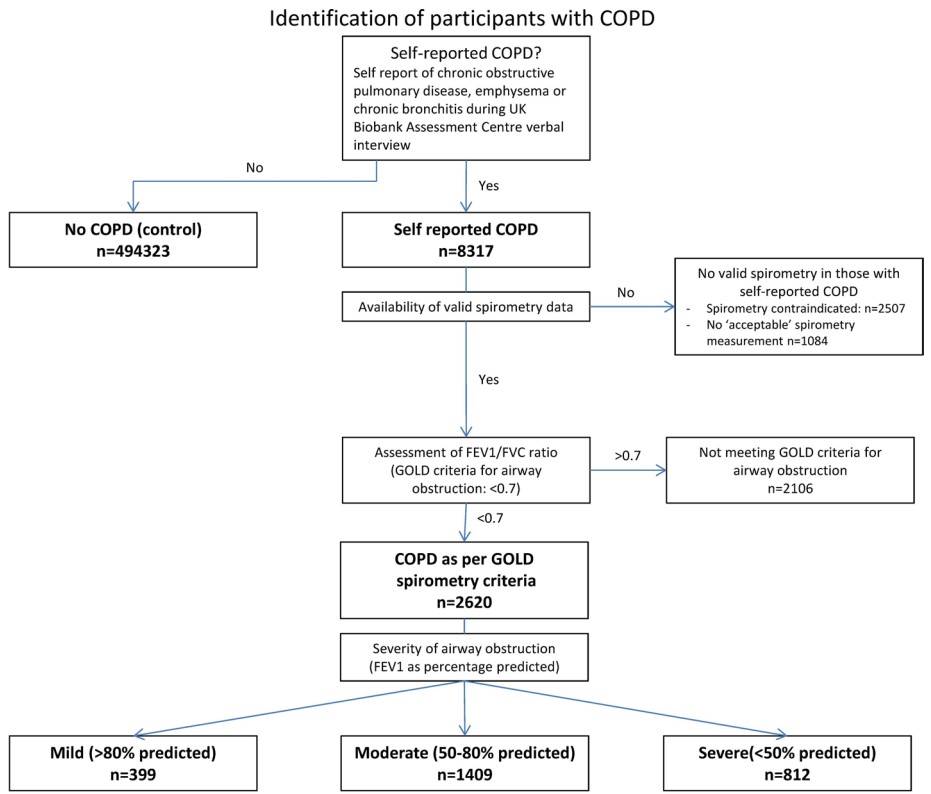

**Figure 1** Flow diagram of identification of participants with 'self-report COPD' and 'GOLD COPD'. COPD, chronic obstructive pulmonary disease; FEV1, forced expiratory volume in 1 s; FVC, forced vital capacity; GOLD, Global Initiative for Chronic Obstructive Lung Disease.

**Table 2** Long-term conditions in those with and without COPD

| | Control n=494 323 count (%) | Self-report COPD n=8317 count (%) | GOLD COPD | | | |
| --- | --- | --- | --- | --- | --- | --- |
| | | | All n=2620 count (%) | Mild n=399 count (%) | Moderate n=1409 count (%) | Severe n=812 count (%) |
| **Total comorbidities (excluding COPD)** | | | | | | |
| ≥4 | 19 959 (4.0) | 1389 (16.7)** | 331 (12.6)** | 46 (11.5) | 191 (13.5) | 94 (11.6) |
| **Total number of medications** | | | | | | |
| ≥1 | 356 406 (72.1) | 7670 (92.2)** | 2452 (93.6)** | 352 (88.2) | 1321 (93.8) | 779 (95.9) |
| ≥5 | 87 286 (17.7) | 4312 (51.8)** | 1349 (51.5)** | 171 (42.9) | 702 (49.8) | 476 (58.6) |
| ≥10 | 10 678 (2.2) | 1269 (15.3)** | 329 (12.6)** | 31 (7.8) | 172 (12.2) | 126 (15.5) |
| **Prevalence of comorbidities** | | | | | | |
| Cardiovascular | 152 891 (30.9) | 3957 (47.6)** | 1156 (44.1)** | 142 (35.6) | 611 (43.4) | 403 (49.6) |
| Hypertension | 130 119 (26.3) | 3206 (38.5)** | 916 (35.0)** | 112 (28.1) | 483 (34.3) | 321 (39.5) |
| CHD | 21 560 (4.4) | 1171 (14.1)** | 315 (12.0) ** | 31 (7.6) | 185 (13.1) | 99 (12.2) |
| Diabetes | 24 737 (5.0) | 766 (9.1)** | 189 (7.2)** | 16 (4.0) | 109 (7.7) | 64 (7.9) |
| Stroke/TIA | 8459 (1.7) | 395 (4.7)** | 98 (3.7)** | 11 (2.8) | 51 (3.6) | 36 (4.4) |
| AF | 3552 (0.7) | 99 (1.2)** | 34 (1.3)** | 3 (0.8) | 16 (1.1) | 15 (1.8) |
| Heart failure | 768 (0.2) | 35 (0.4)** | 6 (0.2) | 0 | 1 (0.1) | 5 (0.6) |
| **Respiratory** | | | | | | |
| Asthma | 55 245 (11.2) | 3048 (36.6)** | 984 (37.6)** | 142 (35) | 523 (37.1) | 319 (39.3) |
| Pulmonary embolus | 4354 (0.9) | 264 (3.2)** | 65 (2.5)** | 12 (3.0) | 32 (2.2) | 21 (2.5) |
| Bronchiectasis | 968 (0.2) | 167 (2.0)** | 39 (1.5)** | 7 (1.8) | 17 (1.2) | 15 (1.8) |
| Pulmonary fib | 504 (0.1) | 67 (0.8)** | 18 (0.7)** | 3 (0.8) | 12 (0.9) | 3 (0.4) |
| Cancer | 37 686 (7.6) | 937 (11.3) | 272 (10.4) | 47 (11.8) | 146 (10.4) | 79 (9.7) |
| Lung | 405 (0.1) | 52 (0.6)** | 15 (0.6)** | 0 | 7 (0.5) | 8 (1.0) |
| Breast | 11 311 (2.3) | 210 (2.5)* | 57 (2.2) | 12 (3.0) | 30 (2.1) | 15 (1.8) |
| Prostate | 3588 (0.7) | 105 (1.3)** | 30 (1.1)* | 5 (1.3) | 12 (0.9) | 13 (1.6) |
| Gastrointestinal | 2925 (0.6) | 96 (1.2)** | 34 (1.3)** | 6 (1.5) | 19 (1.3) | 9 (1.1) |
| Haem | 6170 (1.2) | 124 (1.5)* | 34 (1.3) | 5 (1.3) | 17 (1.2) | 12 (1.5) |
| Gastrointestinal | 55 635 (11.5) | 1737 (20.9)** | 468 (17.9)** | 76 (19.0) | 254 (18.0) | 138 (17.0) |
| Dyspepsia | 37 819 (7.7) | 1257 (15.1)** | 348 (13.3)** | 53 (13.3) | 189 (13.4) | 106 (13.1) |
| Diverticular disease | 5181 (1.0) | 224 (2.7)** | 54 (2.1)** | 6 (1.5) | 32 (2.3) | 16 (2.0) |
| IBS | 11 203 (2.3) | 291 (3.5)** | 64 (2.4)** | 17 (4.3) | 35 (2.5) | 12 (1.5) |
| CLD | 935 (0.2) | 36 (0.4)** | 10 (0.4)* | 2 (0.5) | 10 (0.7) | 3 (0.4) |
| Mental health | 35 822 (7.2) | 1127 (13.6)** | 304 (11.6)** | 54 (13.5) | 162 (11.5) | 88 (10.8) |
| Depression | 27 578 (5.6) | 901 (10.8)** | 233 (8.9)** | 42 (10.5) | 128 (9.1) | 63 (7.8) |
| Anxiety | 8781 (1.8) | 245 (2.9)** | 69 (2.6)** | 13 (3.3) | 36 (2.6) | 20 (2.5) |
| Schizophrenia / bipolar | 1918 (0.4) | 79 (0.9)** | 27 (1.0)** | 3 (0.7) | 15 (1.1) | 9 (1.1) |
| **Other** | | | | | | |
| Other painful | 81 733 (16.5) | 2259 (27.2)** | 655 (25.0)** | 115 (28.8) | 367 (26.0) | 173 (21.3) |
| Osteoporosis | 7700 (1.6) | 342 (4.1)** | 128 (4.9) ** | 21 (5.3) | 67 (4.8) | 40 (4.9) |
| Connective tissue disease | 10 642 (2.2) | 391 (4.7)** | 112 (4.3)** | 19 (4.8) | 72 (5.1) | 21 (2.6) |

Compared with control ($\chi^2$): *P<0.05; **P<0.001.

AF, atrial fibrillation; CHD, coronary heart disease; CLD, chronic liver disease; COPD, chronic obstructive pulmonary disease; GOLD, Global Initiative for Chronic Obstructive Lung Disease; IBS, irritable bowel syndrome; TIA, transient ischaemic attack.

those with COPD plus mental health conditions were most likely to be taking medications predisposing to constipation, CNS depression and bleeding.[52] Within each category of LTC, those with COPD were more likely to be taking multiple medications with similar ADRs than those without. These associations between

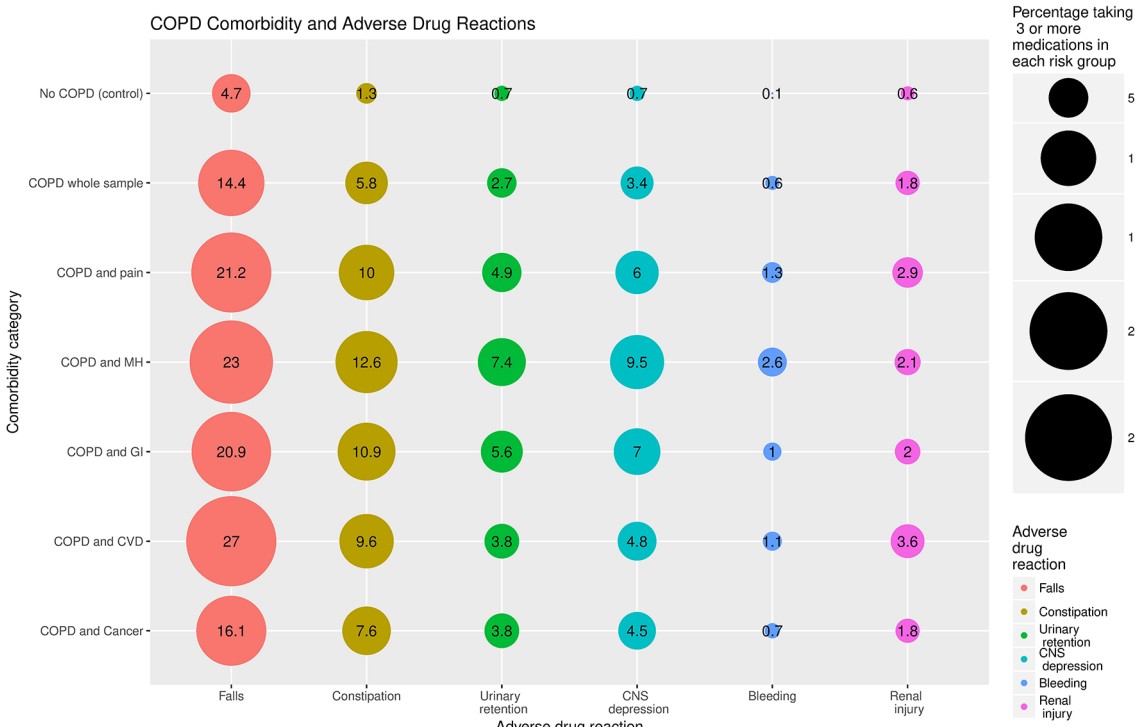

**Figure 2** Bubble plot showing percentage of participants in each comorbidity category taking ≥3 concomitant medications associated with specific adverse drug reactions (ADRs). The size of each bubble represents the percentage of participants in each comorbidity group taking ≥3 concomitant medications associated with specific ADRs according to the Scottish Government Polypharmacy Guideline. CNS, central nervous system; COPD, chronic obstructive pulmonary disease; CVD, cardiovascular disease; GI, gastrointestinal disease; MH, mental health conditions.

patterns of multimorbidity and specific ADR risks have not been described or quantified previously.

### Strengths and limitations

Strengths of this study include the large sample size with representation from different areas of the UK. The range of data collected at UK Biobank assessment centres meant it was possible to compare a range of sociodemographic characteristics as well as spirometry data, the latter being unusual for a large community-based cohort. It is recognised, however, that UK Biobank participants show some evidence of 'healthy volunteer bias', differing from the UK average on a number of socioeconomic, lifestyle and health-related measures. Specifically they are less socioeconomically deprived, less likely to smoke, to be obese and have fewer self-reported health conditions.[53] All LTC diagnoses as well as medication data were

**Table 3** ORs (with 95% CI) for taking three of more medications associated with similar ADRs

| ADR | Self-report COPD compared with no COPD n=502 640 | | GOLD COPD compared with no COPD n=496 943 | |
| | Model 1 n=502 013 | Model 2 n=487 718 | Model 1 n=496 943 | Model 2 n=482 378 |
| | OR (95% CI) | OR (95% CI) | OR (95% CI) | OR (95% CI) |
|---|---|---|---|---|
| Falls | 2.27 (2.13 to 2.42)*** | 1.83 (1.71 to 1.96)*** | 1.66 (1.47 to 1.87)*** | 1.49 (1.30 to 1.69)*** |
| Constipation | 2.71 (2.54 to 2.89)*** | 2.66 (2.39 to 2.96)*** | 2.18 (1.77 to 2.64)*** | 1.82 (1.47 to 2.24)*** |
| Urinary retention | 3.38 (2.94 to 3.87)*** | 2.59 (2.22 to 3.0)*** | 1.98 (1.44 to 2.64)*** | 1.64 (1.18 to 2.21)** |
| CNS depression | 3.75 (3.31 to 4.25)*** | 2.81 (2.45 to 3.22)*** | 2.29 (1.73 to 2.95)*** | 1.87 (1.40 to 2.43)*** |
| Bleeding | 4.60 (3.35 to 6.19)*** | 3.39 (2.40 to 4.66)*** | 2.63 (1.25 to 4.80)** | 1.76 (0.75 to 3.48)† |
| Renal injury | 2.22 (1.86 to 2.62)*** | 1.84 (1.53 to 2.19)*** | 1.94 (1.41 to 2.58)*** | 1.84 (1.33 to 2.49)*** |

Model 1: Adjusted for age, sex and socioeconomic status.
Model 2: Adjusted for age, sex, socioeconomic status, smoking, alcohol frequency, body mass index and physical activity.
**P<0.01; ***P<0.001.
†P>0.05
ADR, adverse drug reaction; CNS, central nervous system; COPD, chronic obstructive pulmonary disease; GOLD, Global Initiative for Chronic Obstructive Lung Disease.

self-reported, with no alternative means of verification. We attempted to minimise this limitation by identifying a subset of those with COPD meeting the GOLD diagnostic criteria and repeating the analyses with this subset. Importantly, spirometry values were also prebronchodilator, which is in contravention to guidelines for diagnosing COPD. Additionally, information was not available about the strength of indication for medications and individual susceptibility to risk, which is a limitation when considering the risk of ADRs.

The use of the Scottish Government Polypharmacy Guideline allowed analysis of potential ADR risk by specific common ADRs. The intended purpose of this guideline, however, was not to identify potential risk from a population sample, but rather to identify potential causes of symptoms or complications. The analysis in this study, therefore, serves only as an approximation of potential risk, not an absolute marker of inappropriate polypharmacy. The cross-sectional nature of this analysis also precludes an analysis of actual harm as a result of polypharmacy. Many of the potential ADRs, such as falls and fractures and renal injury and frequently multifactorial events and may not be directly attributable to medication use. Despite these limitations, however, the co-prescription of multiple medications with similar ADRs strongly implies greater potential for harm. The association of such prescribing patterns with COPD, across a range of potential ADRs, is clear from our findings. This analysis is, to the authors' knowledge, the first to attempt to quantify this risk for specific ADRs in this way.

### Context and implications

The increased prevalence of individual LTCs such as coronary heart disease, hypertension, diabetes, dyspepsia, osteoporosis, cancer, depression and anxiety in those with COPD is similar to the findings from other population-based studies of multimorbidity in COPD.[5 11 54–56] Our finding that cardiovascular disease prevalence increased with increasing severity of COPD is in keeping with the body of literature on cardiovascular disease and COPD, in which high prevalence has been observed in (usually older) cohorts with severe airflow limitation.[5 21] Greater polypharmacy with greater severity of COPD has also been observed previously in older COPD populations,[43 57] although such analyses have been smaller (n=1859 and 398, respectively) and have not assessed the specific patterns of prescribing in COPD. To the best of our knowledge, no previous studies have assessed the risk of ADRs as a result of polypharmacy in COPD. A recent population-based analysis of prescribing data from 310 000 adults in Scotland showed that over 15 years from 1995 to 2010 the proportion of people with polypharmacy and with potentially serious drug-to-drug interactions increased dramatically.[37] The number of prescribed medications was also associated with increased risk of interactions. Our analysis differs in approach from this

analysis, by seeking to identify patterns of prescribing increasing risk of specific adverse events, rather than counting total potential interactions. The strength of our approach lies in highlighting specific patterns of multimorbidity in which specific ADRs are more likely. Our findings can therefore be applied to clinical practice, highlighting the importance of recognising multimorbidity in COPD and being alert to specific ADRs when prescribing medication.

Our findings indicate that in those with COPD the potential for ADRs as a result of combinations of medications is high and this appears to be the result of a high prevalence of extra-pulmonary LTCs. Clinical guidelines for COPD should place greater emphasis on the need for assessment of associated multimorbidity and the risk of associated ADRs. While our analysis shows potential areas where ADR risk exists in COPD (eg, falls or renal injury in those with concomitant cardiovascular disease, CNS depression, bleeding or constipation with concomitant mental health conditions), future research is merited to assess what actual harm could be attributed to such prescribing patterns.

### CONCLUSION

Among UK Biobank participants with COPD, there was considerable multimorbidity and polypharmacy. Those with COPD were highly likely to be concurrently prescribed multiple medications with similar potential adverse effects. Medications contributing to this risk were largely indicated for the management of the associated morbidities rather than COPD. Future research should examine the effects on healthcare outcomes of co-prescribing of multiple drugs with similar potential of ADRs. Clinical guidelines for COPD should emphasise the need for assessment of multimorbidity and the risk of associated ADRs.

**Acknowledgements** This research has been conducted using the UK Biobank Resource, approved project number 14151.

**Contributors** All authors were involved in the conceptualisation and design of the project and interpretation of results. PH carried out the analysis with support from BDJ, RM and BIN. DL provided statistical support. All authors had access to the data. PH wrote the first draft of the paper and all authors commented on subsequent drafts. All authors approved the final draft for publication. FM is the guarantor.

**Funding** This study was supported by a CSO Catalyst Grant CGA/16/39. BDJ was funded by an NHS Research for Scotland career research fellowship.

**Competing interests** None declared.

**Ethics approval** Participants provided full informed consent to participate in UK Biobank, and this study was covered by the generic ethical approval for UK Biobank studies from the NHS National Research Ethics Service (Ref 16/NW/0274).

**Provenance and peer review** Not commissioned; externally peer reviewed.

**Data sharing statement** UK Biobank data is available via www.ukbiobank.ac.uk. Syntax for the generation of derived variables and for the analysis used for this study will be submitted to UK Biobank for record.

provided the original work is properly cited. See: http://creativecommons.org/licenses/by/4.0/

© Article author(s) (or their employer(s) unless otherwise stated in the text of the article) 2018. All rights reserved. No commercial use is permitted unless otherwise expressly granted.

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
