## [Reviewer comments · BMJ Open]

ARTICLE DETAILS

TITLE (PROVISIONAL)	Examining Patterns of Multimorbidity, Polypharmacy and Risk of Adverse Drug Reactions in Chronic Obstructive Pulmonary Disease: A Cross-Sectional UK Biobank Study
AUTHORS	Hanlon, Peter; Nicholl, Barbara; Jani, Bhautesh; McQueenie, Ross; Lee, Duncan; Gallacher, Katie I; Mair, Frances

VERSION 1 – REVIEW

REVIEWER	Lucy Doos University of Birmingham, UK
REVIEW RETURNED	17-Jul-2017

GENERAL COMMENTS	This is really an interesting piece of work in an important area of research particularly with an ageing population. I felt comorbidity rather than multimorbidity fits better in your manuscript as the index disease is COPD. It also helps to avoid the confusion between the two terms. Can you please make the changes to have it as comorbidity all through the manuscript including the title. You need to justify in the introduction section why you chose COPD as disease and look at any other comorbidity with it. You need to add a sentence in the methods section to describe the UK Biobank data to those who are not familiar with it. I wonder if you used any reference for the levels of physical activity. In the methods section, defining comorbidities: Can you please explain what you meant by painful conditions and how you considered it as a comorbidity. What level of pain? what type of pain? this is really very subjective. I understand you had a list in the appendix but it is selective and not sure why only these painful conditions. You mentioned that falls and fractures will be counted as ADR, how can you be sure that these are not from reasons other than polypharmacy? Some data in Table 1 are not in line with those categories in the first column e.g. Alcohol, deprivation. Please consider this. In the result section 8317 reported having COPD out of how many? In the results, you mentioned that those with comorbidity of mental health were more likely to have CNS depression. How can you prove that this is an ADR to medication rather not for the nature of their disease comorbidity? I can't understand how renal injury can be an ADR as mentioned in the discussion section first paragraph, line 306
---

REVIEWER	Federica Sganga UCSC Rome, Italy Dementia, Parkinson, Depression in elderly and Polypharmacy
-----------------	--

REVIEW RETURNED	22-Sep-2017
GENERAL COMMENTS	ACCEPT This manuscript by Hanlon et al. entitled “Examining Patterns of Multimorbidity, Polypharmacy and Risk of Adverse Drug Reactions in Chronic Obstructive Pulmonary Disease: A Cross-Sectional UK Biobank Study” is a potentially interesting study. The study is well designed, methods and results are clearly and tables are very well done. In particular table are very clear and show an important increment of polypharmacy and the increment of ADR in these patients.

VERSION 1 – AUTHOR RESPONSE

Reviewer 1:

We thank the reviewer for her comments that the work is “interesting” and “important”. We have made alterations to the manuscript to address the specific comments raised, and these are detailed below.

Comment 1:

“I felt comorbidity rather than multimorbidity fits better in your manuscript as the index disease is COPD. It also helps to avoid the confusion between the two terms. Can you please make the changes to have it as comorbidity all through the manuscript including the title.”

Authors’ response:

Thank you for this comment. We recognise the need to avoid confusion between the terms ‘comorbidity’ and ‘multimorbidity’, but respectfully request to continue to use the term multimorbidity for the following reasons:

1. We would argue that this is the correct term to use when considering the presence of multiple long term conditions (including the presence of 4 or more long term conditions) as we do in this paper and that use of the term comorbidity would be misleading.
2. Removing the term multimorbidity from the title and manuscript will also make it more difficult for systematic reviewers seeking to synthesise the literature on multimorbidity to identify our paper and cite it within any planned future reviews.

This latter kind of work is a growth area. As authors of this paper are undertaking such a review and have previously undertaken such reviews (1) we can confirm that removal of the multimorbidity term is problematic for future research in this sphere. We are therefore seeking permission to keep the title and manuscript text references to multimorbidity. We believe that our clear definitions of comorbidity and multimorbidity within the introduction should aid clarity. We have previously published other papers on the subject of multimorbidity in the context of a specified index condition and have been allowed to maintain the multimorbidity term.(2) We have attempted to minimise use of the term “comorbidity” to make the paper a clearer read (see changes to manuscript below).

References

1. France, E.F., Wyke, S. , Gunn, J.N., Mair, F. , McLean, G. , and Mercer, S.W. (2012) Multimorbidity in primary care: a systematic review of prospective cohort studies. *British Journal of General Practice*, 62 (597). e297-e307. ISSN 0960-1643 (doi:10.3399/bjgp12X636146)
2. Jani, B.D., Nicholl, B.I., McQueenie, R., Connelly, D., Hanlon, P., Gallacher, K.I., Lee, D. and Mair, F.S. (2017) Multimorbidity and comorbidity in atrial fibrillation and effects on survival: findings from UK Biobank Cohort. *Europeace*. (Accepted for publication: <http://eprints.gla.ac.uk/147770/>)

Changes to manuscript:

To avoid confusion between the terms 'multimorbidity' and 'comorbidity', we have changed the term 'comorbidity' to 'long-term condition' or 'concomitant long-term condition' in the following places:

Lines: 76, 81-86, 98, 106, 160, 161, 162, 165, 219, 222, 228, 259, 263, 266, Table 2, page 15, 275, 282, 284, 286, 287, 299, 303, 320, 323, 355, 357, 359, 371, 372, 376, 378, 379, 380, 386, 389.

Comment 2:

"You need to justify in the introduction section why you chose COPD as disease and look at any other comorbidity with it."

Author's response:

Thank you for this comment. By performing this analysis, we sought to address the problem that although multimorbidity is common, guidelines are generally disease specific and do not offer specific guidance for the management and implications of multimorbidity within the context of any specific index condition. By taking an index condition, examining multimorbidity and exploring the impact of specific additional morbidities on patterns of polypharmacy (and potential associated risk) our aim was to better understand the implications of multimorbidity and different patterns of multimorbidity on prescribing and potential prescribing risks.

While such issues are not unique to COPD, we believe COPD to be a useful exemplar condition on which to base our analysis. This was for several reasons: it is a very common condition; known to be associated with numerous other long term conditions which are wide ranging; and multimorbidity has been demonstrated to have a clinically significant impact on COPD. We present some of the literature behind these justifications in paragraph one of the introduction, and have added the sentences below to clarify and justify our decision to apply specific questions around polypharmacy to COPD as an index condition.

Changes to manuscript:

"The importance of considering the impact of comorbidities in the management of long-term conditions is increasingly recognised, however an immature evidence base means that disease specific guidelines often lack specific recommendations with respect to multimorbidity.⁽³⁴⁾ The prevalence and prognostic significance of multimorbidity in COPD make it a potentially useful exemplar condition in which to consider the specific implications of different patterns of multimorbidity. Polypharmacy is one such implication."

Page 4, Lines 81 to 86

Note reference 34 has been added to reference the NICE multimorbidity guideline (NG56)

Comment 3:

"You need to add a sentence in the methods section to describe the UK Biobank data to those who are not familiar with it."

Authors' response:

Thank you for this, we agree and have added the following to the opening paragraph of the methods section which described the data collection:

Changes to manuscript:

"The UK Biobank is a large, population cohort that recruited voluntary participants from throughout the United Kingdom. Between 2006 and 2010, UK Biobank recruited 502,640 participants aged 37 to 73.

Participants underwent baseline assessments at one of 22 assessment centres throughout England, Scotland and Wales...”

Page 6, Line 111-112

Comment 4:

“I wonder if you used any reference for the levels of physical activity.”

Authors’ response:

Thank you for highlighting that this needs clarification. The data on physical activity, like the other variables used, are based on the questionnaires administered at the UK Biobank assessment centre. To our knowledge, the use of the specific questions used to classify physical activity has not been described in published literature to date. Our classification into four categories (none, low, medium and high) was based on consensus following discussion between the study authors. We have added a link to the UK Biobank data showcase to the manuscript which shows the source of the variables used, as well as altering our text to make clear that the classification into levels was based on our own judgement.

Changes to manuscript:

“Physical activity was self-reported based on a questionnaire administered in the UK Biobank assessment centre (available at <http://biobank.ctsu.ox.ac.uk/crystal/field.cgi?id=6164>). We classified the responses into: none (no physical activity in the last four weeks), low (light ‘DIY’ activity only in the last four weeks), medium (heavy DIY and/or walking for pleasure and/or other exercises in the last four weeks), and high (strenuous sports in the last four weeks).”

Page, Lines 119-124

Comment 5:

“In the methods section, defining comorbidities: Can you please explain what you meant by painful conditions and how you considered it as a comorbidity. What level of pain? what type of pain? this is really very subjective. I understand you had a list in the appendix but it is selective and not sure why only these painful conditions.”

Authors’ response:

Thank you for this comment. We accept that considering pain as a long term condition is subjective, and, thank the reviewer for highlighting the need for clarification around this issue. The overall list of long term conditions that we included was based on prior literature in the field of multimorbidity research:

“The list of included long term conditions was taken from a list of 42 morbidities originally established for a large epidemiological study of multimorbidity in Scotland, through systematic review, the Quality and Outcomes Framework, NHS Scotland and an expert panel (48), and subsequently amended for UK Biobank (49).”

Page 8, Lines 161-164

The inclusion of ‘painful conditions’ as a long term condition was in line with this existing literature. In studies such as Barnett et al referenced above, painful conditions were defined by the prescription of analgesics. Equivalent data were not available in our dataset, and in the context of polypharmacy we felt it would be inappropriate to define a condition by the use of specific medication. We therefore took an alternative approach to group common conditions for which pain is a predominant feature as

'painful conditions'. This list was agreed upon by discussion between two of the study authors (BN and FM). We have altered the text to clarify this:

Changes to manuscript:

"The inclusion of 'other painful conditions' comprised long-term conditions in which pain is a predominant feature (particularly as this is likely to influence medication use). It should be noted that such a list is not exhaustive, but intended to cover common conditions frequently requiring prescription of analgesics (e.g. osteoarthritis, back pain, headaches etc.) ."

Page 8, Lines 164-168

Authors' response, continued:

We also thank the reviewer for highlighting the 'selective' nature of the list of painful conditions, and potential exclusion of some conditions of which pain is a feature. We accept that the list of 'other painful conditions' is not exhaustive and have sought to clarify this in above. This is an important point, particularly in the subsequent analysis in which patterns of medication use are compared between those with COPD and different categories of long term conditions. After considering the reviewers comments we have expanded the category of 'painful conditions' used in the logistic regression models to include all conditions within 'other painful conditions' plus the 'connective tissue diseases' group. This comprises rheumatoid arthritis as well as other inflammatory arthropathies (full list in appendix 1). The list of painful conditions used in our analysis, while not exhaustive, is improved by including these important omissions. For clarity we have renamed this group 'painful conditions/inflammatory arthropathies'.

Changes to manuscript:

The following changes have been made to address this point:

"Morbidity were categorised for the purposes of this analysis into cardiovascular disease, gastrointestinal disease, mental health conditions, cancer, and painful conditions/inflammatory arthropathies (comprising the list of 'other painful conditions' mentioned above, plus connective tissue diseases). Full details of conditions comprising each category can be found in appendix 1. "

Page 8, Lines 168-172

The percentage of participants with COPD plus a painful condition taking 3 or more medications in each risk group has been recalculated based on the updated list of conditions, and figure 2 has been updated accordingly.

The adjusted analyses showing the association between COPD and painful conditions (appendix 3, table S1), as well as the subgroup analyses of participants with painful conditions (appendix 3, table S8) have been re-analysed based on the updated list. Appendix 3 has been amended to include the results of the updated analysis.

Comment 6:

"You mentioned that falls and fractures will be counted as ADR, how can you be sure that these are not from reasons other than polypharmacy?"

Authors' response

Thank you. We agree with the reviewer that several events, such as falls and fractures, can be due to a number of potential causes and not just polypharmacy. By defining those on 3 or more medications with the same potential ADR as 'at risk' we do not seek to claim that all events are by definition

caused by such polypharmacy, but rather seek to identify a group in whom the risk of a specific event associated with polypharmacy is higher.

The choice of specific ADRs included in our analysis was based on the Scottish Government Model of Care Polypharmacy Working Group: Polypharmacy Guidance. This guideline lists falls and fractures as one such ADR.

The reviewer rightly points out that some of these events, such as falls and fractures, are multifactorial events and difficult to attribute directly to polypharmacy with certainty. As the polypharmacy guideline acknowledges, however, these are events to which a patient's susceptibility can be increased by taking multiple medications that increase the risk of an event.

Changes to manuscript:

We have added the following to the methods section describing the ADR definitions to highlight the reviewer's point that events (such as falls and fractures) may not be directly attributable to polypharmacy.

"We identified six potential ADRs (falls/fractures, constipation, urinary retention, CNS depression, bleeding and renal injury) for which the proportion of participants taking three or more associated medications could be assessed. It should be noted that several of these event (e.g. falls/fractures, CNS depression) are often multifactorial, and medication may be a contributing factor rather than a definitive cause. As the guideline acknowledges, however, these are clinical events of which the risk is increased by taking multiple associated medications."

Page 9, Lines 185-190

We have also added the following sentence to the limitations section of the discussion to further highlight this issue:

"Many of the potential ADRs, such as falls and fractures and renal injury, and frequently multifactorial events and may not be directly attributable to medication use."

Page 19, Lines 346-348

Comment 7:

"Some data in Table 1 are not in line with those categories in the first column e.g. Alcohol, deprivation. Please consider this."

Authors' response

We thank the reviewer for these comments and have corrected the alignment in table 1 (page 13)

Comment 8:

"In the result section 8317 reported having COPD out of how many? "

Authors' response:

Thank you. All participants in the cohort were asked to report any health conditions that were applicable to them. The denominator on which we based our analysis was 502,619.

Changes to manuscript:

We have added this to the results section to clarify:

“At the time of recruitment, 8317 out of 502,619 participants reported having COPD (1.7%) and are referred to here as the self-report COPD group.”

Page 12, Line 242

Comment 9:

“In the results, you mentioned that those with comorbidity of mental health were more likely to have CNS depression. How can you prove that this is an ADR to medication rather not for the nature of their disease comorbidity?”

Authors' response:

Thank you for this comment. We agree with the reviewer that CNS depression, particularly in this context, may be due to a number of causes. We do not seek to prove that any specific event is directly caused by medication (indeed our analysis is not designed to do so), but rather show what patterns of multimorbidity are associated with taking multiple medications with similar risks.

We have sought to address this specific issue more comprehensively in our response to comment 6. In addition to this, we have made the following changes to the results section to clarify the specific section highlighted by the reviewer:

Changes to manuscript:

“Participants with COPD plus mental health conditions had the highest percentages taking three or more medications with a risk of constipation, CNS depression or bleeding.”

Page 16, Lines 289-291

We have also amended the following section of the discussion:

“Those with COPD plus cardiovascular disease were most likely to be taking multiple medications associated with increased risk of falls or renal injury, while those with COPD plus mental health conditions were most likely to be taking medications predisposing to constipation, CNS depression and bleeding.(50)”

Page 18, Lines 320-323

Comment 10:

“I can't understand how renal injury can be an ADR as mentioned in the discussion section first paragraph, line 306”

Authors' response:

Thank you. As we discuss at greater length in response to comment 6, we acknowledge that renal injury is a multifactorial event. While medications are not the cause in every case, taking multiple associated medications has been shown to increase the risk of such events (Lapi et al 2013).

Reference:

Lapi F, Azoulay L, Yin H, Nessim SJ, Suissa S. Concurrent use of diuretics, angiotensin converting enzyme inhibitors, and angiotensin receptor blockers with non-steroidal anti-inflammatory drugs and risk of acute kidney injury: nested case-control study. *Bmj*. 2013 Jan 8;346:e8525.

Changes to manuscript:

We have adjusted the wording of the section highlighted by the reviewer to clarify the distinction:

“Those with COPD plus cardiovascular disease were most likely to be taking multiple medications associated with increased risk of falls or renal injury,”

Page 18, Lines 230-322

We have also made the following adjustment to the results section

“Participants with COPD plus cardiovascular disease had the highest percentage taking three or more medications with a risk of falls or renal injury.”

Page 16, Lines 288-289

Reviewer 2

We thank the reviewer for their comments that “the study is well designed, methods and results are clearly and tables are very well done. In particular table are very clear and show an important increment of polypharmacy and the increment of ADR in these patients.”

VERSION 2 – REVIEW

REVIEWER	Lucy Doos University of Birmingham
REVIEW RETURNED	21-Nov-2017

GENERAL COMMENTS	This is a clearly written manuscript in an important area of research. However, the authors didn't refer to the increasingly importance of research in this area in an ageing society. A couple of sentences will add to the strength and importance of the study. The authors need to do the following changes to improve the manuscript: -In the Strength and limitations section: Please review the language in the first bullet point-Page 7 Line 121: move the link to the references and add the citation-Page 9 Line 173: Please add reference for BNF-Page 10 Line 192 under statistical analysis: please review the English language as the sentence isn't clear as it is.-Table 2 prevalence of comorbidity: I am not sure how DVT is linked to respiratory diseases if it is not PE. Please rerun the analysis for PE only-Page 19 Line 314: more than half reported polypharmacy: please add the actual percentage-Reference 34 (NICE): authors need to add the link and date of access to it.
--

VERSION 2 – AUTHOR RESPONSE

Reviewer 1:

Reviewer comment:

This is a clearly written manuscript in an important area of research. However, the authors didn't refer to the increasingly importance of research in this area in an ageing society. A couple of sentences will add to the strength and importance of the study.

Author response:

Thank you for this comment. We agree with the importance of highlighting the relevance of changing societal demographics on multimorbidity and polypharmacy and have added the following two sentences to emphasise this:

“The importance of considering the impact of multimorbidity in the management of long-term conditions is increasingly recognised, particularly in the context of an ageing society in which the prevalence of multimorbidity is growing.(34)” Page 4 lines 81-83

“Polypharmacy has been associated with increased risk of adverse drug reactions (ADRs)(37-39) and potentially preventable hospital admissions, particularly in the elderly.(40, 41) This is particularly pertinent in an ageing society, in which the a rising prevalence of polypharmacy has been observed.(34, 37)” Pages 4-5, lines 90-93

Further reviewer comments and responses below:

The authors need to do the following changes to improve the manuscript:

-In the Strength and limitations section: Please review the language in the first bullet point

Thank you for highlighting the grammatical error in this bullet point. It now reads:

“This paper assesses multimorbidity, polypharmacy and risk of adverse drug reactions in UK Biobank participants with self-reported COPD compared with those without COPD.”

-Page 7 Line 121: move the link to the references and add the citation

We have added this link to the reference section as a citation. (reference 45)

-Page 9 Line 173: Please add reference for BNF

We have added a reference for the BNF here (reference 51)

-Page 10 Line 192 under statistical analysis: please review the English language as the sentence isn't clear as it is.

Thank you. We have amended this sentence, which now reads:

“All analyses were planned prior to inspection of the data.”

-Table 2 prevalence of comorbidity: I am not sure how DVT is linked to respiratory diseases if it is not PE. Please rerun the analysis for PE only

We have repeated this part of the analysis and now display values for PE only

-Page 19 Line 314: more than half reported polypharmacy: please add the actual percentage

We have added the percentage (52%) to this section

-Reference 34 (NICE): authors need to add the link and date of access to it.

Thank you. We have added the URL and date of access to the reference in question.

VERSION 3 – REVIEW

REVIEWER	Dr Lucy Doos University of Birmingham, UK
REVIEW RETURNED	29-Nov-2017
GENERAL COMMENTS	The manuscript looks much better now as the authors made all the amendments as requested